# The Application of Robust Least Squares Method in Frequency Lock Loop Fusion for Global Navigation Satellite System Receivers

**DOI:** 10.3390/s20041224

**Published:** 2020-02-23

**Authors:** Mengyue Han, Qian Wang, Yuanlan Wen, Min He, Xiufeng He

**Affiliations:** School of Earth Sciences and Engineering, Hohai University, Nanjing 210098, China; 15951936312@163.com (M.H.); wwwgwyl@126.com (Y.W.); mhe@hhu.edu.cn (M.H.); xfhe@hhu.edu.cn (X.H.)

**Keywords:** satellite navigation, signal tracking, VFLL, loop segmentation, robust least squares

## Abstract

The tracking accuracy of a traditional Frequency Lock Loop (FLL) decreases significantly in a complex environment, thus reducing the overall performance of a satellite receiver. In order to ensure high tracking accuracy of a receiver in a complex environment, this paper proposes a new tracking loop combining the vector FLL (VFLL) with a robust least squares method, which accurately matches the weights of received signals of different qualities to ensure high positioning accuracy. The weights of received signals are selected at the signal level, not at the observation level. In this paper, the ranges of strong and weak signals of the loop are determined according to the different expressions of the distribution function at different signal strengths, and the concept of loop segmentation is introduced. The segmentation results of the FLL are taken as a basis of the weight selection, and then combined with the Institute of Geodesy and Geophysics (IGGIII) weight function to obtain the equivalent weight matrix; the experiments are conducted to prove the advantages of the proposed method over the traditional methods. The experimental results show that the proposed VFLL tracking method has strong denoising capability under both normal- signal and harsh application environment conditions. Accordingly, the proposed model has a promising application perspective.

## 1. Introduction

The Global Satellite Navigation System (GNSS) plays an important role in the economic and military fields. The GNSS represents a space-based radio navigation and positioning system that provides users with information on their three-dimensional coordinates, speed, and the time when a satellite signal is received at any location on the Earth’s surface or in near-Earth space. At present, China is developing the BeiDou Navigation Satellite System (BDS-3). With the more intense application of GNSS in military and civil fields, higher requirements for reliability and environmental adaptability of the GNSS have been put forward. As an important part of a receiving system, the Frequency Lock Loop (FLL) has been widely used in the receiver tracking loop due to its good robustness and dynamic performance. The least squares method is one of the basic methods for geographic data processing. The robust least squares method denotes a robust weighting method that was proposed to address the phenomenon of gross error existence in observed data. Therefore, the FLL can be combined with the robust least squares method to design a more accurate tracking loop to give different weights to satellites according to the received signal strength, so as to improve the overall positioning accuracy. 

Most studies on the FLL have been focused on improving a certain part of the FLL structure to improve the FLL performance, or adopting the FLL to assist other loops to improve the overall performance of a tracking loop. The traditional tracking loop has been mostly implemented in the form of a scalar loop, but the vector tracking loop has been widely used in a GNSS receiver because of its strong dynamic ability to re-capture weak signals when the lock is lost [1]. Different from the traditional scalar loop, the vector tracking loop makes full use of the correlation between the channels, channels are not independent of each other, and the vector output of each discriminator represents the measurement of the tracking loop, such as code phase, carrier phase and carrier frequency. Based on the code phase, carrier phase and carrier frequency, the user’s motion status, clock difference, clock drift and other information used in the positioning process can be estimated. In [2], the concept of vector tracking called Vector Delay Lock Loop (VDLL), was proposed for the first time. The Vector Frequency Lock Loop (VFLL) is another form of a vector tracking loop, which was first proposed by Pany in 2005 [1]. In [3], the model of VFLL was established, the performance of the VFLL was evaluated by the simulation, and the advantages of the VFLL algorithm were verified experimentally. In [4], the authors proposed a joint estimation method of multi-channel parameters in the signal domain based on the least squares method, aiming at achieving an optimal carrier-to-noise ratio (CNR), and obtaining the parameters of the pseudo-code tracking and carrier tracking loops in the full channel range. In [5], a VFLL assisted Phase Lock Loop (PLL) model was proposed, the system model and feedback generation process were described, and the feasibility of the newly proposed model in a scintillation scenario was tested. A new type of double differential Vector Phase Locked Loop (VPLL) was introduced in [6]; the proposed loop improves the sensitivity of the original VFLL. Further, a new high-precision vector tracking algorithm was proposed in [7]. This algorithm adopts the VFLL assisting the PLL to replace the traditional FLL assisting the PLL, thus improving the accuracy of carrier phase observation and velocity observation.

The robust least squares method represents a robust weighted method initially proposed for data processing. The robust feature means maintaining good stability and ability to resist signal shelter and interference, where stability feature means that when there is a certain difference between a model on which the estimation method is based and the actual model, this method can still obtain relatively stable results. On the other hand, the ability to resist interference means that the results obtained by this method are not significantly affected by a small gross error present in the observation sample. Many studies have been conducted on its application and statistical testing methods, and it has been compared with other methods. At present, the robust least squares method is mostly used for data processing.

In 1980, a robust estimation theory was introduced into the geomatics field by Krarup and Kubik, and the famous Danish method was proposed in [8]. Besides, in [9], the authors proposed a robust weighted total least squares method, which took into account the error of the coefficient matrix observation. In [10], according to the quantitative relationship between the absolute value of observation residuals and the observation error, the observation data were divided into three categories: normal observation values with no gross error, suspicious observation values with possible gross error, and obsolete observation values that denote a gross error. A correlated robust estimation solution similar to the correlated least-squares estimation was constructed in [11] by using the relevant equivalence weight principle, and it was named the Institute of Geodesy and Geophysics (IGG) scheme. In [12], two independent noise statistical estimation strategies were used for sequential adaptation of parameters, and a robust adaptive Kalman carrier tracking loop was designed to achieve desirable performance gain. Lastly, a real-time adaptive weighting model was proposed in [13] to reduce the observation error of the field-specific unmodeled pseudo-code. Moreover, statistical testing is an important part of the robust least square method. For instance, the Detection, Identification and Adaptation (DIA) method represents a robust least squares method. In [14], a unifying framework that captures the combined estimation was introduced, and the scheme of the DIA method was tested. As one of characteristics of the DIA method, the paper proved that the estimator was unbiased under H_0_, but not under any of the alternative hypotheses. A hybrid method termed as the iterative least squares method with the initial values constrained by using the Fukuda-Johnson (F-J) method proposed in [15], and the inversion effect of this method was verified by Chi-square test [16].

At present, there are a few studies on the FLL and least square method fusion, but most of them studied the two methods separately. Most studies on the FLL focus on improving the FLL structure to improve the tracking performance, such as improving the FLL discriminator or filter. The least square method is commonly used for e data processing, and the positioning solution is found after the observation data are obtained. In order to obtain better observation data in complex environments, the data processing method is fused with the FLL, and the data are screened during signal tracking. Therefore, based on the FLL theory, this paper integrates the VFLL and the robust least squares to construct a more accurate tracking loop. 

The main contributions of this paper are as follows: (1)Using the difference in the thermal noise distribution function at different signal strengths, the strong- and weak-signal ranges of the FLL are determined, and the concept of the loop segmentation is introduced.(2)According to the segmentation results of the fusion of FLL and the robust least squares method, a VFLL based on the robust least squares method is constructed and verified by the experiment.

The rest of the paper is organized as follows. In Section 2, the four-quadrant arctangent discriminator of the FLL is used to derive the output errors of the discriminator and filter; also, according to the different distribution characteristics of the distribution function at a different signal strength, the loop error was segmented to determine whether the signal-to-noise ratio (SNR) and when is high or low. In Section 3, the FLL segmentation results are fused with the robust least squares method to improve the tracking performance of the original VFLL model, and an improved VFLL model is constructed. In Section 4, the accuracy of the VFLL based on the robust least square method is verified by the experiment, and its performance is compared with those of the traditional Scalar Frequency Lock Loop (SFLL), the original VFLL, and the weighting method at the observation level. The comparison results show that the VFLL based on the robust least squares method has certain advantages over the other methods in complex environments, thus improving the tracking accuracy of the original FLL. In Section 5, the conclusions are given.

## 2. FLL Segmentation Model

The FLL represents a form of carrier loop, and it has been widely used in the receiver tracking loop due to its good robustness and dynamic performance. The FLL structure mainly includes three parts: frequency discriminator, loop filter and Numerically Controlled Oscillator (NCO). The purpose of the FLL is to keep the same frequency of the replicated carrier and the received carrier. First, the frequency difference between the replicated carrier and the received carrier is identified by the frequency discriminator, and then the frequency of the replicated carrier, which represents the output of the NCO carrier is adjusted accordingly. After several iterations of frequency adjusting, the frequencies of the replicated and received carriers finally achieves dynamic consistency.

In this paper, we consider only the errors caused by environmental thermal noise because this noise is a dominant part of the overall error. The main idea is to use the probability density function *P(ϕ)* to calculate the variance of the loop output, and then divide the loop into three parts: strong signal, general signal and weak signal according to the distribution characteristics of *P*(*ϕ*) under different signal intensities. The function *P*(*ϕ*) can be expressed as [17]:(1)$$P\left(\varphi \right)=\frac{e^{-\frac{SNR}{2}}}{2\pi }\left(1+\sqrt{\frac{\pi SNR}{2}}\cos\left(\varphi \right)e^{\frac{SNR\cos\varphi^{2}}{2}}•\left(1+erf\left\{\frac{SNR\cos\varphi }{\sqrt{2}}\right\}\right)\right),-\pi <\varphi <\pi$$
where φ denotes the phase difference, erf{⋅} represents the Gaussian error function. The distribution of function given by Equation (1) is closely related to the SNR, as shown in Figure 1.

In Figure 1, the SNR represents the signal strength. As the signal strength decreases, the function *P*(*ϕ*) changes from a Gaussian distribution to a uniform distribution gradually [18]. At different signal strength, there are different mathematical distribution characteristics. Therefore, the characteristics of this function can be used to determine the range of strong and weak signals of the FLL.

### 2.1. Frequency Discriminator Output Error

The main function of the FLL discriminator is to output the frequency difference between the replicated carrier and the received carrier. In the selection of a discriminator, the influence of the data bit transition represents an important factor, so it should be considered. However, a division discriminator can eliminate this influence. Therefore, a four-quadrant arctangent discriminator, which has a larger linear working range, is usually used in the FLL [17]. 

The estimated value of the frequency error denotes a difference between the phase estimations at time *m*-1 and time *m* [19], and the frequency residual of a signal interfered by the thermal noise can be estimated by [17]:(2)$$e_{m}\approx \frac{K_{D}}{T_{L}}\left(\theta_{m}-\theta_{m-1}\right)+n_{m}^{\omega }$$
where *K_D_* denotes the gain determined by both the discriminator function and the SNR, *T_L_* denotes the integration time, $n_{m}^{\omega }$ represents the thermal noise, and lastly, $\theta_{m}$ and $\theta_{m-1}$ denote the instantaneous carrier phases at time *m* and time *m*−1, respectively.

The error function of a four-quadrant arctangent (*A*tan2) discriminator is expressed as [17]:(3)$$e_{m}^{A\tan2}=\frac{1}{T_{L}}\arctan(Dot,Cross)$$
where $D\mathrm{ot}=I_{m}I_{m-1}+Q_{m}Q_{m-1}$, and $Cross=Q_{m}I_{m-1}-I_{m}Q_{m-1}$. The in-phase and quadrature components of the input signal at two adjacent time moments are expressed as $I_{m}=\cos\left(\delta_{\omega }T_{L}+\phi_{2}\right)$, $I_{m-1}=\cos\phi_{1}$, $Q_{m}=\sin\left(\delta_{\omega }T_{L}+\phi_{2}\right)$, and $Q_{m-1}=\sin\phi_{1}$, respectively.

According to the probability density function P(φ), the variance of the output error of a discriminator is given by [17]:(4)$$Var\left(n^{\omega }\right)=\frac{1}{T_{L}^{2}}{\int_{-\pi }^{\pi }\int_{-\pi }^{\pi }f\left(\phi_{2}-\phi_{1}\right)}^{2}P\left(\phi_{1}\right)P\left(\phi_{2}\right)d\phi_{1}d\phi_{2}$$
where f(⋅) denotes the discriminator function of a loop. In this work, the four-quadrant arctangent discriminator function is expressed as [17]:(5)$$f^{Atan2}\left(\delta_{\omega }T_{L}\right)=\delta_{\omega }T_{L}\left(u\left(\delta_{\omega }T_{L}+\pi \right)-u\left(\delta_{\omega }T_{L}-\pi \right)\right)$$
where u(x) denotes a unit step function, and $u\left(x\right)=\left\{ \begin{array}{l} 0,x<0\\ 1,x\geq 0 \end{array}\right.$; $\delta_{\omega }$ denotes the carrier frequency error, and $T_{L}$ represents the integration time; lastly, $\delta_{\omega }T_{L}$ denotes the phase increment in interval $T_{L}$.

The value of $K_{D}$ affects the FLL performance significantly. After simplification [17], the frequency discrimination gain $K_{D}$ can be expressed as [17]:(6)$$K_{D}^{A\tan2}=1-4\pi \int_{-\pi }^{\pi }P\left(\theta \right)P\left(\theta -\pi \right)d\theta$$

The distribution of function P(φ) is complex, which makes it difficult to obtain an analytical solution for error $Var\left(n^{\omega }\right)$ in Equation (4). Therefore, the approximate method proposed in [17] is adopted to obtain the numerical solution to Equation (4). At high SNR, function P(φ) obeys the Gaussian distribution, so its variance can be expressed by [20]:(7)$$Var\left(n^{\omega }\right)=\frac{2}{SNRT_{L}^{2}}(rad/s^{2})$$

With a decrease in the SNR, the distribution of function P(φ) begins to differ from the Gaussian distribution, and at an extremely low SNR, function P(φ) follows the uniform distribution in the range [−π,π] [20], and its variance tends to $\frac{\pi^{2}}{3T_{L}^{2}}$ [20], which represents an upper bound of the theoretical value of the solution to Equation (4). In practice, the FLL has a certain tracking threshold. In order to simulate an actual situation, in this work, the tracking threshold value is considered as an upper bound value of the solution to Equation (4). When the error threshold is reached, the SNR is considered to be low, which is given by [21,22]:(8)$$3Var(n^{\omega })\leq \frac{1}{4T_{L}}$$

The numerical solution to the output variance of a discriminator that is given by Equation (4) is obtained by [17]:(9)$$Var(n^{\omega })\approx \frac{\frac{2}{SNR}\left(1-e^{-c_{1}SNR^{2}}\right)+\frac{\pi^{2}}{3}e^{-c_{2}SNR}}{T_{L}^{2}}$$
where coefficients $c_{1}=0.0524$ and $\;c_{2}=0.50301$ denote the results of the weighted least square fitting of SNR in the range (−3 dB, 30 dB), and the range of this signal is applicable to the GNSS receiver [17].

There are two segmentation points of the discriminator output, one is the point where the variance starts to differ from the so-called high-SNR situation, and the other denotes the point where the variance starts to approach the low-SNR situation, which means that the loop may lose lock. The purpose of finding the segmentation points is to determine whether the SNR is high or low. Combining Equations (7) and (9), and Equations (8) and (9), we get the required segmentation points; the carrier-to-noise ratio (CNR) is taken as a unit, and CNR = SNR/*T*, where *T* denotes the integration time. After this conversion, the following results are obtained:(10)$$CNR_{1}=39.320\;\mathrm{dB}-\mathrm{Hz}$$
(11)$$CNR_{2}=48.519\;\mathrm{dB}-\mathrm{Hz}$$

### 2.2. Filter Output Error

After the signal passes through the discriminator, it is fed to the filter. The second-order loop filter is the most commonly used in a tracking loop. The output error of the second-order loop filter is given by [17]:(12)$$\begin{array}{ll} \sigma_{\delta \omega }^{2}= & \frac{\left(R_{0}^{n}+2R_{1}^{n})T_{L}(2A_{0}^{2}K_{D}+A_{1}(2+A_{0}K_{D}T_{L}\right)}{A_{0}K_{D}^{2}(4-K_{D}T_{L}(2A_{0}+A_{1}T_{L}))}\\ & -\frac{2R_{1}^{n}K_{D}T_{L}^{2}(A_{1}^{2}T_{L}+A_{0}K_{D}(A_{0}+A_{1}T_{L})(2A_{0}+A_{1}T_{L})}{A_{0}K_{D}^{2}(4-K_{D}T_{L}(2A_{0}+A_{1}T_{L}))} \end{array}$$
where $A_{0}$ and $A_{1}$ denote the filter coefficients, $K_{D}$ denotes the frequency discriminator gain, $T_{L}$ denotes the integration time; and, $R_{0}^{n}$ and $R_{1}^{n}$ denote the autocorrelation coefficients, where $R_{0}^{n}$ represents the total discriminator noise power, and it is $Var(n^{\omega })$, while $R_{1}^{n}$ represents the cross-correlation degree of the noise, given an offset of $T_{L}$. There is a certain relationship between $R_{1}^{n}$ and $R_{0}^{n}$. At high SNR, the ratio $-R_{1}^{n}/R_{0}^{n}$ is close to 1/2, as the SNR decreases, the value of $-R_{1}^{n}/R_{0}^{n}$ also decreases, and it becomes zero finally. After the least square fitting process, $R_{1}^{n}$ can be expressed as [17]:(13)$$R_{1}^{n}\approx -\frac{1}{2}(1-e^{-0.4864\cdot SNR})R_{0}^{n}$$

For the second-order loop filter, the two filter coefficients are given by [17]:(14)$$A_{0}=\frac{1-e^{-2\beta }}{K_{D}T_{L}}$$
(15)$$A_{1}=\frac{e^{-2\beta }\left(1-e^{\beta \left(1+\eta \right)}\right)\left(1-e^{\beta \left(1-\eta \right)}\right)}{K_{D}T_{L}^{2}}$$
where β represents the oscillation fading factor, and η denotes the oscillation damping factor. However, the analytic solution to β can be difficulty obtained. Generally, a quadratic polynomial $\beta \approx c_{1}B_{\omega }T_{L}-c_{2}{\left(B_{\omega }T_{L}\right)}^{2}$ is used for approximation of the analytical solution to β, and solutions to $c_{1}$ and $c_{2}$ are obtained by the least squares fitting method; namely, $c_{1}=\frac{\pi }{4}$ and $c_{2}=\frac{1}{6}$ [17], where $B_{\omega }$ denotes the noise bandwidth, and $T_{L}$ represents the integral time. The damping coefficient is expressed as $\eta^{2}=-1$, so the best damping ratio is expressed as $\xi =\frac{1}{\sqrt{2}}$ [22].

After the coefficients of the loop filter are determined, Equation (12) can be solved. According to the expression of $R_{0}^{n}$ at different SNR values and the relationship between $R_{0}^{n}$ and $R_{1}^{n}$, by substituting $R_{0}^{n}$ and $R_{1}^{n}$ into Equation (12), Equation (12) can also be used to calculate the segmentation points. At a high SNR, $R_{0}^{n}$, that is $Var(n^{\omega })$ in Equation (4), satisfies the Gaussian distribution, and the variance is $\frac{2}{SNRT_{L}^{2}}$, and at a low SNR, the tracking threshold $R_{0}^{n}=\frac{1}{12T}$ still denotes the upper bound of Equation (4). Substituting different values of $R_{0}^{n}$ into Equation (12), the segmentation points can be calculated, and the filter output error can be summarized as a segmented function, which is given by:(16)$$\sigma_{\delta \omega }^{2}=\left\{ \begin{array}{l} \frac{2c_{1}+\left(2c_{1}-c_{2}\right)\left(e^{-0.4864SNR}-1\right)}{c_{3}K_{D}^{2}T_{L}^{2}SNR}\;\mathrm{CNR}>48.496\;\mathrm{dB}-\mathrm{Hz}\\ \frac{c_{1}\frac{1}{{\left(12T\right)}^{2}}+\frac{1}{{\left(12T\right)}^{2}}\left(2c_{1}-c_{2}\right)\left(e^{-0.4864SNR}-1\right)}{c_{3}K_{D}^{2}T_{L}^{2}}\;\mathrm{CNR}<25.079\;\mathrm{dB}-\mathrm{Hz}\; \end{array}\right.$$
where:(17)$$c_{1}=T_{L}(2A_{0}^{2}K_{D}^{2}+2A_{1}K_{D}+A_{0}A_{1}K_{D}^{2}T_{L})$$
(18)$$c_{2}=2T_{L}^{2}\left(A_{1}^{2}K_{D}^{2}T_{L}+\left(A_{0}^{2}K_{D}^{2}+A_{0}A_{1}K_{D}^{2}T_{L}\right)\left(2A_{0}K_{D}+A_{1}K_{D}T_{L}\right)\right)$$
(19)$$c_{3}=A_{0}K_{D}\left(4-T_{L}\left(2A_{0}K_{D}+A_{1}K_{D}T_{L}\right)\right)$$

## 3. Improved VFLL Model Based on Robust Least Squares Method 

In the satellite weighting method, commonly used weighting models are mostly based on the satellite elevation angle and SNR value [23]. The VFLL based on the robust least squares method is to weigh the satellite according to the SNR value. However, the weight is determined based on the signal strength in the tracking stage, rather than in the data processing stage, in order to obtain better observation data. 

The two main parts of the improved VFLL model based on the robust least square method are a discriminator and a filter. In the discriminator, the least square method is used to determine the required parameters for tracking. Then the obtained residuals are fed back to the receiving channel through the Kalman filter. By constructing the equivalent weight matrix, the frequency value is calculated accurately by the Kalman filter. The structure diagram of the proposed model is shown in Figure 2.

In the Figure 2, each channel is used to track a satellite, and *δα*, *δτ*, and *δφ* respectively denote differences in the signal amplitude, code phase, and carrier phase between the replicated carrier and the received carrier. The least square method is used to calculate the values of *δα*, *δτ* and *δφ*. Then, these values are used to calculate precisely the carrier frequency difference, *δf*, by the Kalman filter. In the Kalman filter, the equivalent weight matrix is constructed by Equation (28), which is provided in the following.

### 3.1. Discriminator Output

The downlink signal in the satellite navigation receivers represents an input to the baseband filter, which is obtained after frequency conversion by the radio frequency (RF) module and carrier modulation, and it is given by [4]:(20)$$x\left(t\right)=\sum_{i=1}^{M}a_{i}S_{i}\left(t-\tau_{i}\right)\exp\left\{j2\pi f_{d,i}t+j\varphi_{i}\right\}+n\left(t\right)$$
where t denotes the signal receiving time, M represents the number of receiving satellite, $a_{i}$ is the signal amplitude, $\tau_{i}$ is the code phase delay, $\varphi_{i}$ is the carrier phase and $f_{d,i}$ is the carrier frequency; $S_{i}\left(t-\tau_{i}\right)$ denotes the modulation code of the corresponding signal [4], and n(t) represents the white Gaussian noise. The process of receiver tracking is to obtain the above-mentioned signal parameters i.e., $a_{i}$, $\tau_{i}$, $\varphi_{i}$ and $f_{d,i}$, from the noisy received signal. In the traditional loop, the parameters are obtained by the independent discriminator. However, in the proposed model, these parameters are obtained by the least squares method and Kalman filtering.

Equation (20) expresses a nonlinear function, so the first-order of Taylor expansion can be employed to linearize the non-linear signal. In the proposed model, Taylor expansion is applied to the three parameters, i.e., $a_{i}$, $\tau_{i}$, and $\varphi_{i}$ in order to reduce the algorithm complexity and obtain the carrier frequency value more accurately because the carrier frequency $f_{d,i}$ can be calculated more accurately using these three parameters. Then, Equation (20) can be expanded as follows:(21)$$x\left(t\right)=\hat{x}{\left(t\right)}_{i}+\frac{\partial x}{\partial a_{i}}\delta a+\frac{\partial x}{\partial \tau_{i}}\delta \tau +\frac{\partial x}{\partial \varphi_{i}}\delta \varphi +n_{i}$$

Equation (21) expresses the signal linearization result of a channel *i*. For multiple channels, Equation (21) can be expressed in the matrix form, which is given by:(22)$$\left[\begin{array}{lll} \frac{\partial X}{\partial a_{1}} & \frac{\partial X}{\partial \tau_{1}} & \frac{\partial X}{\partial \varphi_{1}}\\ \frac{\partial X}{\partial a_{2}} & \frac{\partial X}{\partial \tau_{2}} & \frac{\partial X}{\partial \varphi_{2}}\;\\ \ldots .. & & \\ \frac{\partial X}{\partial a_{M}} & \frac{\partial X}{\partial \tau_{M}} & \frac{\partial X}{\partial \varphi_{M}}\; \end{array}\right]\left[\begin{array}{l} \delta a\\ \delta \tau \\ \delta \varphi \end{array}\right]=\left[\begin{array}{l} X_{1}-{\hat{X}}_{1}\\ X_{2}-{\hat{X}}_{2}\\ \ldots ..\\ X_{M}-{\hat{X}}_{M} \end{array}\right]-\left[\begin{array}{l} N_{1}\\ N_{2}\\ \ldots .\\ N_{M} \end{array}\right]$$
(23)$$A\delta \theta =X-\hat{X}-N$$
where X represents the received baseband signal vector, $\hat{X}$ denotes local replica of the received baseband signal vector, N represents the white Gaussian noise vector, A is the least squares coefficient matrix, and lastly, δθ denotes the synchronization parameter residual vector [4]. By applying the least squares method, the solution to δθ can be obtained by:(24)$$\delta \theta ={\left(A^{T}A\right)}^{-1}A^{T}(X-\hat{X})={\left(A^{T}A\right)}^{-1}A^{T}\Delta x$$
where Δx denotes the difference between X and $\hat{X}$. The satellite synchronization parameter δθ is obtained in the discriminator stage, after which follows the filtering stage. The proposed model adopts Kalman Filter (KF), and in the filtering stage, parameters $a_{i}$, $\tau_{i}$, and $\varphi_{i}$ are used to determine the value of $f_{d,i}$ accurately, and robust estimation is performed. The weight function is used to determine the weight of each channel while obtaining the value of $f_{d,i}$. This process is described in detail in the following.

### 3.2. Filter Output 

The state vector and measurement vector of the KF system are respectively expressed as [4]:(25)$$\begin{array}{l} S_{n}={\left[a_{n}^{T}\tau_{n}^{T}\varphi_{n}^{T}f_{n}^{T}\right]}^{T}\\ \delta \theta =\left[\delta a_{n}^{T}\delta \tau_{n}^{T}\delta \varphi_{n}^{T}\right] \end{array}$$
where $a_{n}^{T}$ denotes the signal amplitude vector, $\tau_{n}^{T}$ represents the code phase delay vector, $\varphi_{n}^{T}$ denotes the carrier phase vector, and $f_{n}^{T}$ denotes the carrier frequency vector. The KF state transition equation is expressed as:(26)$$\begin{array}{l} S_{n+1}=\left[\begin{array}{l} a_{n+1}\\ \tau_{n+1}\\ \varphi_{n+1}\\ f_{n+1} \end{array}\right]=F\left[\begin{array}{l} a_{n}\\ \tau_{n}\\ \varphi_{n}\\ f_{n} \end{array}\right]+\omega_{n}=FS_{n}+FK_{k}\delta \theta \\ F=\left[\begin{array}{llll} I_{M\times M} & 0 & 0 & 0\\ 0 & I_{M\times M} & 0 & -\frac{T_{c}}{f}I_{M\times M}\\ 0 & 0 & I_{M\times M} & 2\pi TI_{M\times M}\\ 0 & 0 & 0 & I_{M\times M} \end{array}\right] \end{array}$$
where $I_{M\times M}$ denotes the identity matrix, *f* represents the signal frequency, F is the KF coefficient matrix, $K_{k}$ is the KF gain matrix, and $\omega_{n}$ is the process noise matrix; $K_{k}$ is expressed as:(27)$$K_{k}={\bar{P}}_{n}F_{n}^{T}{\left(F{\bar{P}}_{n}F_{n}^{T}+R\right)}^{-1}$$
where ***R*** denotes the noise covariance matrix and in this KF model, it is expressed as ***R = (A^T^A)***^−1^; ${\bar{P}}_{n}$ is the equivalent weight matrix in the proposed model [24], and it is different from that of original KF model. Namely, in the proposed model, ${\bar{P}}_{n}$ denotes the product of the weight factor matrix and the initial weight matrix, which is given by:(28)$${\bar{P}}_{n}=\bar{Y}\cdot {\bar{P}}_{n,0}$$

In Equation (28), the initial weight matrix (${\bar{P}}_{n,0}$) is defined as the identity matrix, and the weight factor matrix ($\bar{Y}$) is obtained by commonly used three-segment weight function, and then robust estimation is performed. The IGGIII weight function represents a classical three-segment weight function, but it is mostly used in the data processing stage, which is given as [25]:(29)$$y_{i}=\left\{ \begin{array}{ll} 1 & \left|{\tilde{v}}_{i}\right|\;<k_{0}\\ \frac{k_{0}}{\left|{\tilde{v}}_{i}\right|}{\left(\frac{k_{1}-\left|{\tilde{v}}_{i}\right|}{k_{1}-k_{0}}\right)}^{2} & k_{0}\leq \left|{\tilde{v}}_{i}\right|<k_{1}\\ 0 & \left|{\tilde{v}}_{i}\right|\geq k_{1} \end{array}\right.$$
where $k_{0}$ and $k_{1}\;$ are constants; $k_{0}$ is in the range 1.0–1.5, and $k_{1}\;$ is in the range 3.0–8.5 [25]; ${\tilde{v}}_{i}$ is the *i*^th^ standardized residual, and it can be calculated by:(30)$${\tilde{v}}_{i}=\frac{v_{i}}{\sigma_{v_{i}}}$$
where $v_{i}$ represents the carrier frequency residual of the *i*th channel, and $\sigma_{v_{i}}$ represents the standard deviation of the *i*^th^ residual.

As given by Equation (29), the IGGIII weight function divides the data into three data types based on the observed value residual, which are good observed value, the observed value requiring weight reduction, and a gross error requiring elimination. However, the expression in Equation (29) requires a known residual value, so it can be used only in the data processing stage. But the implication of dividing the data into three segments is excellent, and the FLL segmentation results can correspond to the implication. So, in order to determine the weights at the signal level, we need to define a weight function according to the segmentation results. 

In Section 2.2, two segmentation points of the filter output are calculated, and the error is divided into three segments as follows. When the CNR value is greater than 48.496 dB-Hz, the input signal is considered as good; when the CNR is between 25.079 dB-Hz and 48.496 dB-Hz, the input signal is the value requiring weight reduction; and lastly, when the CNR is less than 25.079 dB-Hz, the input signal is the value to be eliminated. According to the FLL segmentation results, a weight function is defined to obtain the weight factor linearly at the signal level:(31)$${\bar{y}}_{i}=\left\{ \begin{array}{ll} 1 & CNR_{i}\geq 48.496\;\mathrm{dB}-\mathrm{Hz}\;\\ \frac{CNR_{i}}{48.496} & 25.079\;\mathrm{dB}-\mathrm{Hz}<CNR_{i}<48.496\;\mathrm{dB}-\mathrm{Hz}\;\\ 0 & CNR_{i}\leq 25.079\;\mathrm{dB}-\mathrm{Hz}\; \end{array}\right.$$
where ${\bar{y}}_{i}$ represents the weight factor of *i*^th^ channel and it can form a diagonal matrix which represents the weight factor matrix ($\bar{Y}$) used to obtain the equivalent weight matrix ${\bar{P}}_{n}$ by Equation (28). Thus, when the values in Equations (27) and (26) are known, the value of carrier frequency $f_{n}^{T}$ can be calculated. 

The Kalman filter flow of the proposed model is represented in the Figure 3, where at time k,*δα*, *δτ* and *δφ* are the inputs of the Kalman filter, and ${\bar{P}}_{n}$ and ***R*** are used to calculate the gain matrix in Equation (27). The value of ${\bar{P}}_{n}$ calculated by Equations (28) and (31) is the equivalent weight matrix that relates to the SNR. Next, *δf* can be obtained. Then, at time k+1, the Kalman filter repeats all these steps.

According to the received signal strength and interval partitioning of Equation (31), the input signal can be divided into three categories: good observation values, suspicious observation values, and gross error values. Such a division can help manage the situation of obviously low signal quality effectively and ensure the final result is not affected by the gross error. 

## 4. Experimental Results and Discussion

In order to verify the reliability of the theoretical analysis presented in Section 2 and the advantages of the proposed VFLL model based on the robust least squares method that is introduced in Section 3, the simulations were carried out, and the obtained results are presented in this section. The simulations were divided into four parts. The first part verified the reliability of theoretical analysis of segmentation results of the FLL; the second part verified the performance of the modified VFLL model in open environment, the third part verified the advantages of the modified VFLL model in complex environment; lastly, the fourth part compared the difference between the weights at the observation level and the weights at the signal level. These parts are explained in detail in the following. 

### 4.1. FLL Segmentation Verification 

The segmentation of discriminator output is explained in Section 2.1, whereas the filter output is presented in Section 2.2. In order to verify the theoretical derivation, the receiver designed in MATLAB software was used in the simulations. The integral time was set as 2 ms, which represents an empirical value commonly used in this type of simulation and the dynamic acceleration of the satellite was set to 0. The signal range was 20–50 dB-Hz; this range of signal strength was chosen because it is suitable for GNSS receivers. The output segmentation results of the FLL frequency discriminator are presented in Figure 4. In Figure 4a, the numerical solution to the output error in Equation (9) is verified by satellite simulation data. Figure 4b shows the high SNR linear model obtained by Equation (7), the FLL tracking threshold obtained by Equation (8), and the numerical solution to the error was obtained by Equation (9), In Figure 4, the segmentation points are marked with rectangles and circles. In Figure 4c, the difference between the numerical solution to the error in Equation (9) and the high SNR linear model in Equation (7) is displayed. In Figure 4d, the difference between the numerical solution to the error in Equation (9) and the tracking threshold in Equation (8) is illustrated. In Figure 4c,d, the values of the two segmentation points are marked.

As can be seen in Figure 4a, the simulation results are consistent with the theoretical analysis results. Moreover, as shown in Figure 4c,d, there was a small interval between the high-SNR and low-SNR cases. This was because the discriminator output error was not smoothed, and its value was large, which led to a small difference between the high and low SNR values. Obviously, the error needed to be reduced further by a filter, and that process is explained in the following.

The simulations were also conducted at the loop noise bandwidth of 25 Hz and the integration time of 2 ms. As shown in Figure 5a, the numerical solution to the output error after filtering in Equation (12) was verified by satellite simulation data. Figure 5b shows the numerical solution to the error after filtering obtained by substituting Equation (9) into Equation (12); the high SNR linear model after filtering was obtained by substituting Equation (7) into Equation (12), the FLL tracking threshold was obtained after filtering by substituting Equation (8) into Equation (12), and the simulation data were generated by the receiver designed in MATLAB software, they are all presented in Figure 5. In Figure 5c,d, the difference between the numerical solution to the output error after filtering and the high SNR model and the tracking threshold are presented. All the other simulation parameters and conditions were same as those used to obtain the results presented in Figure 4.

By comparing the results presented in Figure 5a with those presented in Figure 4a, it can be found that the loop output error was significantly reduced after filtering, after filtering, the simulation results were consistent with the theoretical analysis results. In Figure 5c,d, it can be seen that the interval between high and low SNR values was obviously expanded, which ensured avoiding to lose useful information and getting more accurate results. The simulation results in both Figure 4 and Figure 5 show good consistency with the theoretical analysis results.

### 4.2. VFLL Model Performance Verification in Open Environment

The VFLL model based on the robust least squares method is introduced in detail in Section 3. In order to verify the advantages of the proposed model, the receiver designed in MATLAB software was used in the experiment. Namely, in open environment, the quality of satellite signals is generally good, so a design method does not cause damage to the original loop and improves the tracking accuracy to a certain extent. This is why the simulations were conducted under such environmental conditions. Also, we compared the tracking results of the traditional SFLL, VFLL, and VFLL based on the robust least squares method at the CNR of 42 dB-Hz, using a total of 11 receiving channels. Under low dynamic conditions, the signal strength was 42 dB-Hz, and the integral time was 2 ms. The results that correspond to channel 3 (CH3), which was one of 11 channels, are shown in Figure 6, where the tracking results of the three models are presented.

As can be noticed in Figure 6, the errors of the two VFLL models were significantly reduced compared with that of the traditional SFLL, and the tracking error of the original VFLL was larger than that of the VFLL model based on the robust least squares method. The frequency tracking errors of all eleven channels are given in Table 1, where the best results are provided in bold.

The results in Table 1 show that the tracking results of the proposed VFLL model based on the robust least squares method were the best among all the results. Thus, in the open environment, the VFLL model based on the robust least-squares method did not cause damage to the original loop and was more accurate than the other methods. The frequency error of the proposed model was controlled at about 0.2 Hz. The positioning speed accuracy of the three models is presented in Figure 7a, where it is shown that the results of the two VFLL models are obviously better than that of the SFLL model. The clear version of the dotted line in Figure 7a is presented in Figure 7b in order to present the results more clearly, where it is obvious that the accuracy of the results of the proposed VFLL model is further improved compared to the original VFLL model.

As can be seen in Figure 7, the two VFLL models achieved obvious improvements compared to the traditional SFLL. However, the proposed VFLL model based on the robust least squares method was more accurate than the original VFLL. The standard deviations of the velocity error of all the three models are provided in Table 2.

In Table 2, it can be seen that the proposed model achieved the best results among all the models, achieving the velocity error smaller than 0.05 m/s. In order to compare the results of various methods from a statistical perspective, the standard errors of a unit weight of all the methods are shown in Figure 8:

As can be seen from Figure 8, the proposed model has obvious advantages. Compared with the other two models, the standard error of unit weight of SFLL is about 2.5 m/s while that of the VFLL model based on the robust least squares method was about 0.3 m/s.

### 4.3. VFLL Model Performance Verification under Partial Occlusion

In order to verify the advantages of the proposed VFLL model further, the experiments were conducted in the environment with partial occlusion, and the scenario that included the gross error was simulated. The same as in the previous experiments, 11 receiving channels were set, the CNR value of channels 1–5 was set to 50 dB-Hz, corresponding to the good observation value with no gross error, the CNR value of channels 6–9 was set to 40 dB-Hz, corresponding to the suspicious observation value with a possible gross error, and the CNR value of channels 10–11 channel was set to 15 dB-Hz, corresponding to the elimination observation value with a gross error. The other experimental settings were the same as those used in the previously presented experiments, and the tracking conditions of three models were also compared.

The frequency error of the traditional SFLL is shown in Figure 9. As presented in Figure 9, in channels CH10 and CH11, there were obviously gross errors, so the tracking error could not converge, and it had a large deviation, which reflected the disadvantage of the traditional SFLL.

The results of the original VFLL are shown in Figure 10, where it can be seen that this model could resist the coarse errors to a certain extent, but the tracking errors of channels CH10 and CH11 still fluctuated greatly, and the tracking results were not satisfactory. The proposed VFLL model based on the robust least squares method was used for verification. Due to the difference in the signal strength of different channels, in the VFLL model based on the robust least squares method, the weight factor of each channel was different. The weight factors of all the channels were calculated by Equation (31). The channels CH10 and CH11 did not participate in the channel parameter solution. In the experiment, the proposed method eliminated the two channels with a poor signal and tracked only channels with non-coarse signals to ensure accurate results, as shown in Figure 11.

In the experiment, the signal strength of channels 6–9 was 40 dB-Hz. In both the traditional SFLL model and the VFLL models, the tracking result was significantly affected by the low-signal quality satellites, i.e., channels CH10 and CH11. However, this was not the case with the proposed VFLL model. The tracking frequency errors of the three models of CH6 are shown in Figure 12.

As shown in Figure 12, the proposed VFLL ensured that the tracking results of the satellites whose signal quality was in the “middle zone” were not significantly interfered with the gross errors. The tracking frequency errors of all the channels were calculated, and they are given in Table 3.

As presented in Table 3, compared with the two traditional models, the proposed model had a significantly lower error in channels CH6-CH9. The signal in channels CH1-CH5 was good, so they accounted for the majority of the signal, and the overall change in the error of channels CH1-CH5 was not significant.

In order to demonstrate the advantage of the proposed model further, the final positioning results of the models were compared. The positioning error of the traditional SFLL model is presented in Figure 13; the positioning errors of the traditional and proposed VFLL models are presented in Figure 14.

As shown in Figure 13, under the condition of obvious gross error, the SFLL had a large error, and the positioning result was disturbed and could not converge. On the other hand, as presented in Figure 14, the two VFLL models could track the corresponding satellite, but the positioning result of the traditional VFLL model was less accurate than that of the proposed VFLL model that, eliminated the satellite with obvious gross error and ensured high accuracy of the final result. The positioning errors of the three models within 30 s were statistically analyzed, and the results are shown in Table 4.

As shown in Table 4, the proposed VFLL had the best accuracy. In Table 4, it can be seen that in the presence of satellite signals with poor quality, the proposed VFLL had more advantages, ensuring the tracking of satellites with an excellent quality and protecting the satellites with a high signal strength from the interference with the satellites with a low signal strength, thus providing high accuracy of the final positioning results. Under the complex environmental conditions, the proposed VFLL could control the three-dimensional positioning error within 0.3 m, the traditional SFLL could not converge and the positioning error of the traditional VFLL was about 1.17 m; the standard error of a unit weight is shown in Figure 15.

Since the error of the SFLL did not converge, in order to clearly illustrate the advantages of the proposed model, in Figure 15, there are only two kinds of models. In Figure 15, it can be clearly seen that under the condition of partial occlusion, the standard error of a unit weight of the proposed model was controlled at about 2 m while that of the VFLL was about 6 m.

### 4.4. Comparison between Observation- and Signal-Level Weights

Generally, the robust estimation is conducted at the data processing stage, that is, the observation level, while in the proposed VFLL, it is conducted at the signal level of a satellite, whose step is compared in advance to the commonly used robust estimation. The proposed model ensures that the satellites with better signal quality are used to determine the synchronization parameters, so that the synchronization parameters are highly accurate.

In the experiments, when the weight was determined at the observation level, Equation (29) was used to calculate the weight factor. In Equation (29), $k_{0}$ was set to 1, and $k_{1}$ was set to 3, these values of $k_{0}$ and $k_{1}$ were chosen such that to eliminate large residual value and select a better satellite data source. After the simplification, we got:(32)$$y_{i}=\left\{ \begin{array}{ll} 1 & \left|{\tilde{v}}_{i}\right|\;<1\\ \frac{1}{\left|{\tilde{v}}_{i}\right|}{\left(\frac{3-\left|{\tilde{v}}_{i}\right|}{2}\right)}^{2} & \;1\leq \left|{\tilde{v}}_{i}\right|<3\\ 0 & \left|{\tilde{v}}_{i}\right|\geq 3 \end{array}\right.$$
where ${\tilde{v}}_{i}$ was the *i*^th^ standardized residual. In the location calculation stage, the weighted location calculation was conducted using the weight factor given by Equation (32). There are some differences in the results calculated by Equations (31) and (32). The channels where the CNR value was set to 40 dB-Hz or 50 dB-Hz accounted for the same proportion by Equation (32), without any precise distinction. The weights of channel when the CNR value was set to 15 dB-Hz were not zero according to Equation (32). The weighted least-squares method was used for position calculation, and the results were compared with those of the two VFLL models, as shown in Figure 16.

The IGGIII weight function was used to determine the weight of the VFLL in the position calculation. The accuracy of this method was higher than that of the VFLL, because it appropriately reduced the proportion of poor observations. However, the results of the VFLL based on the robust least squares method were more accurate, and robust estimation was carried out when the channel parameters were obtained; thus, the obtained synchronization parameters were more accurate, and the quality of observation data used in the positioning and calculation was higher compared to the other two models in Figure 16. The standard error of a unit weight is shown in Figure 17.

As presented in the Figure 17, that the data processing with robust the least square method improved the positioning accuracy, and it was more effective in signal tracking that the other model. Therefore, the proposed VFLL based on the robust least squares method is feasible and reliable.

### 4.5. Summary

In this Section, the theoretical analysis of the FLL segment presented in Section 2 and the proposed VFLL model introduced in Section 3 is verified by simulation. The results show that the output segmentation results of discriminator and filter are consistent with the experimental results; thus the theoretical analysis is reliable.

The proposed VFLL model was verified from three aspects. First, in the open environment, the proposed model controlled the frequency error of each channel at about 0.2 Hz, and the accuracy of speed was nearly 50 percent better than that of the original VFLL. Next, in the complex environment, the positioning accuracy of the proposed VFLL model was controlled at about 0.3 m, while the traditional SFLL could not converge, and the positioning accuracy of original VFLL was about 1 m. The proposed model was not affected by the gross error in the complex environment, which ensured high final positioning accuracy. Finally, the weight determination method of the proposed model is compared with that of traditional data processing. The results show that the proposed weight determination method on the signal level is superior to the weight determination method on the data processing level.

## 5. Conclusions

In this paper, a new VFLL based on the robust least squares method is proposed. Based on the thermal noise distribution function, the variances of the FLL discriminator and filter outputs are derived to determine the ranges of strong and weak signals corresponding to the loop. Taking the segmentation results of the FLL as a basis of weight selection and combining it with the IGGIII weight function, the weight factor is calculated. The satellites are divided into three categories according to the signal quality, and the VFLL model based on the robust least square method is constructed. The theoretical analysis was verified by simulation, and the simulation results were in good agreement with the theoretical results. Also, the tracking results of the traditional SFLL, traditional VFLL, and proposed VFLL based on the robust least squares method were compared. The results showed that when the signal quality of all the channels was the same, the proposed VFLL did not interfere with the original loop, and the performance of the proposed VFLL was better than those of the traditional SFLL and VFLL. 

In the presence of satellites with poor signal quality, the proposed FLL has the obvious advantages. Due to the precise determination of signal quality, the anti-noise ability in the positioning process is improved, and thus, the positioning accuracy is significantly improved compared with the two traditional methods. By comparing the weighting processes at the observation and signal levels, it can be found that more accurate tracking parameters can be obtained by weighting at the signal level, and the positioning accuracy is also improved compared with that at the observation level. The method proposed in this paper can restore useful information in the navigation signals better than the original methods, so it has a promising application perspective in complex and harsh environments.

## Figures and Tables

**Figure 1 sensors-20-01224-f001:**
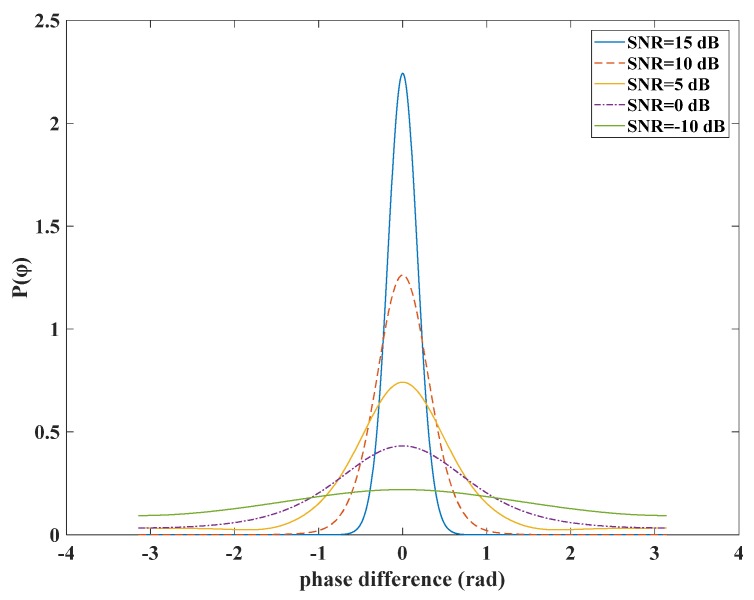
Relationship between the probability density function and SNR.

**Figure 2 sensors-20-01224-f002:**
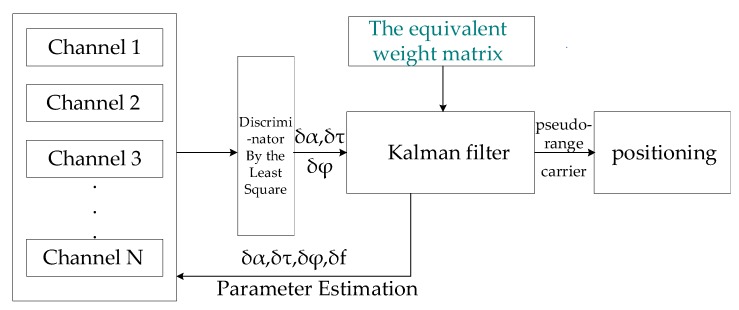
The structure of the proposed model.

**Figure 3 sensors-20-01224-f003:**
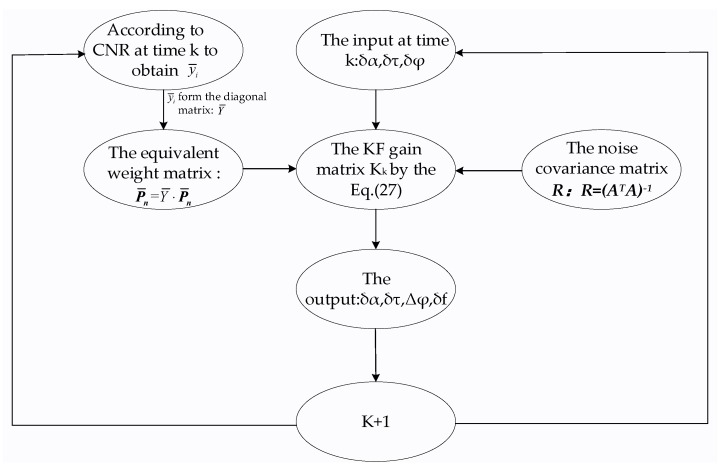
The flow of the Kalman filter.

**Figure 4 sensors-20-01224-f004:**
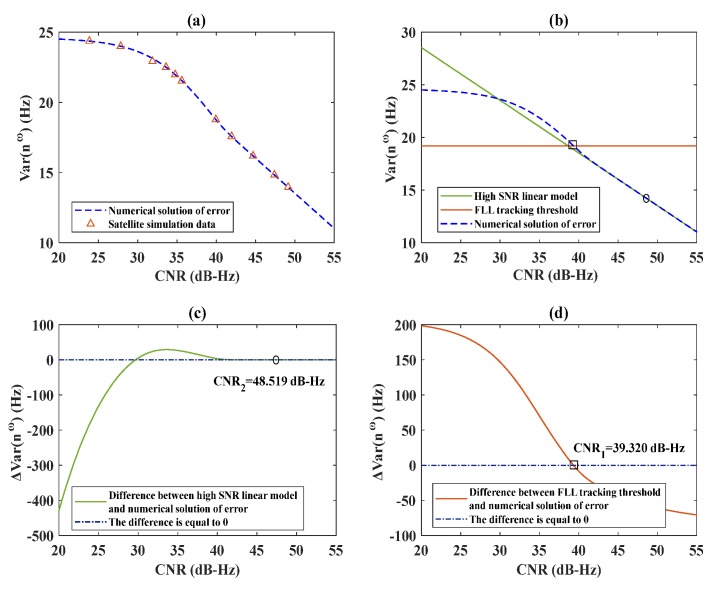
The output segmentation results of the FLL frequency discriminator.

**Figure 5 sensors-20-01224-f005:**
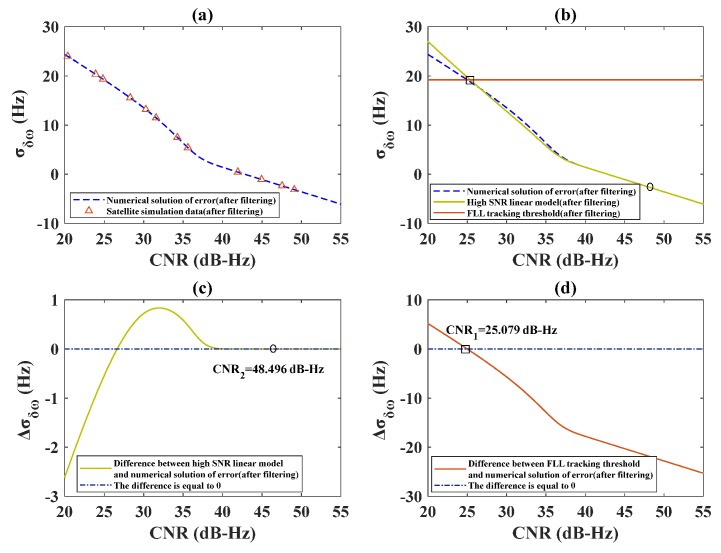
The FLL filter output error dependence on the CNR value.

**Figure 6 sensors-20-01224-f006:**
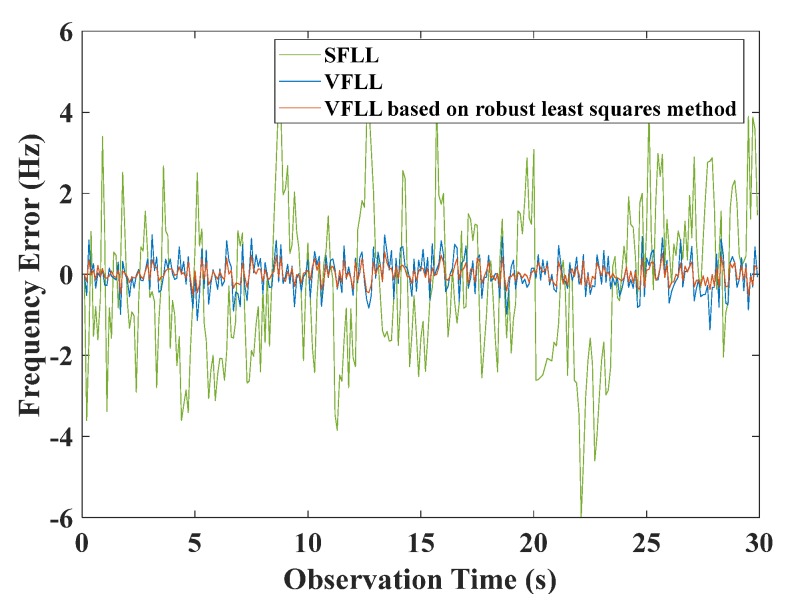
The CH3 tracking frequency errors of the three models under the open-environment conditions.

**Figure 7 sensors-20-01224-f007:**
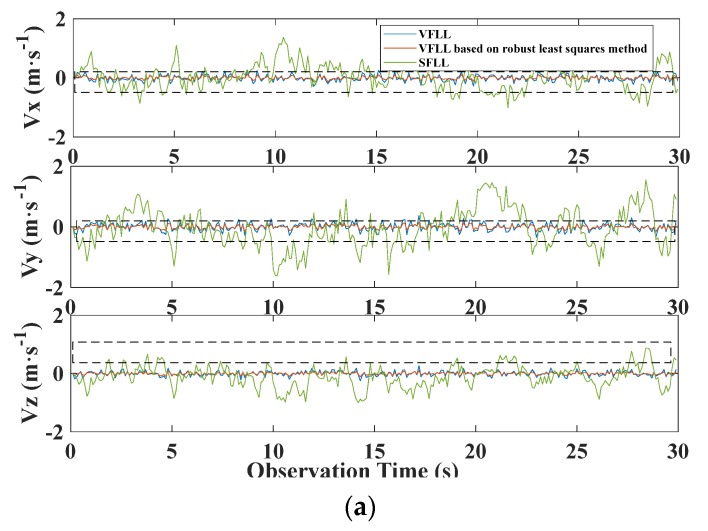
(**a**) Velocity error comparison of the three models. (**b**) Enlarged view of the results presented in the dotted-line bordered region in figure (**a**).

**Figure 8 sensors-20-01224-f008:**
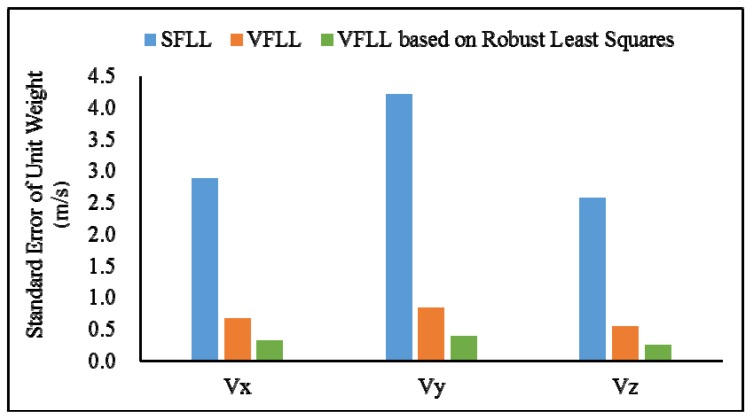
The standard errors of a unit weight of the velocity error of the three models.

**Figure 9 sensors-20-01224-f009:**
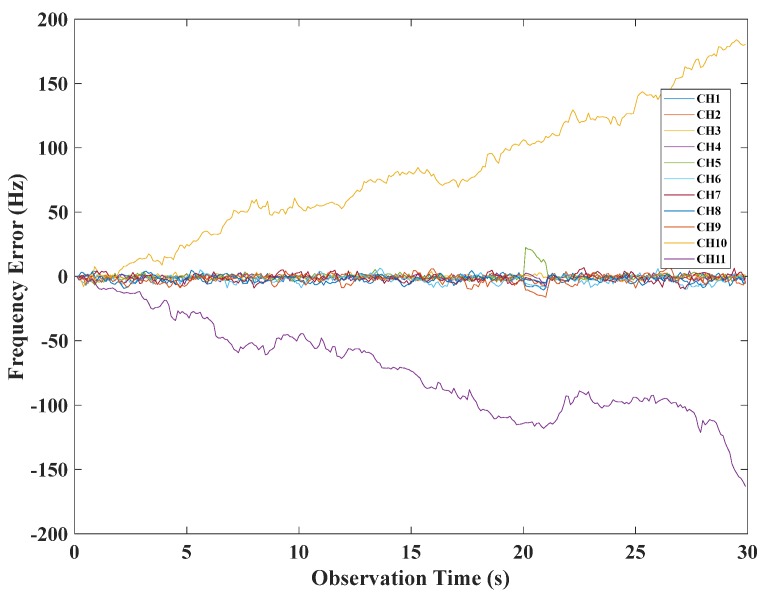
The frequency tracking error of the traditional SFLL.

**Figure 10 sensors-20-01224-f010:**
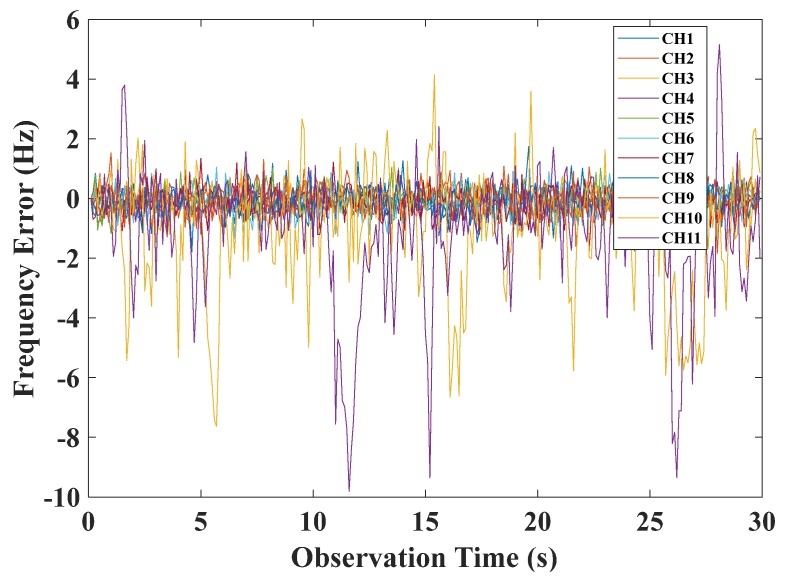
The frequency tracking error of the traditional VFLL.

**Figure 11 sensors-20-01224-f011:**
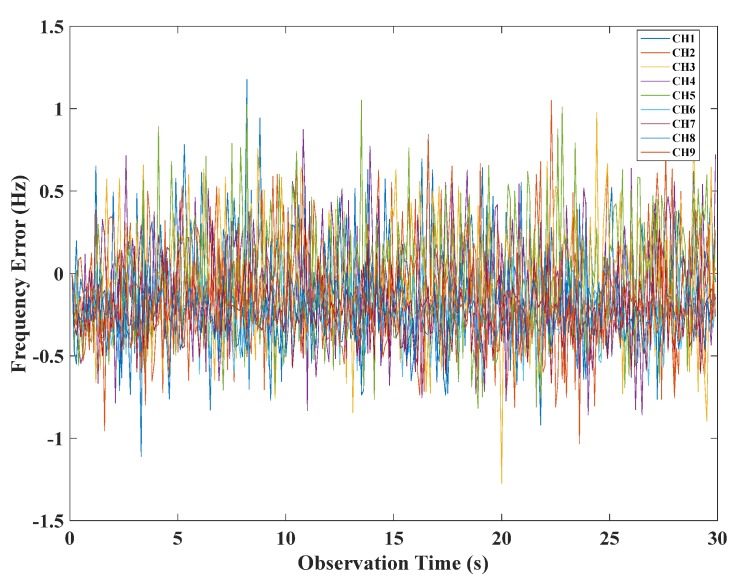
The frequency tracking error of the VFLL model based on the robust least squares method.

**Figure 12 sensors-20-01224-f012:**
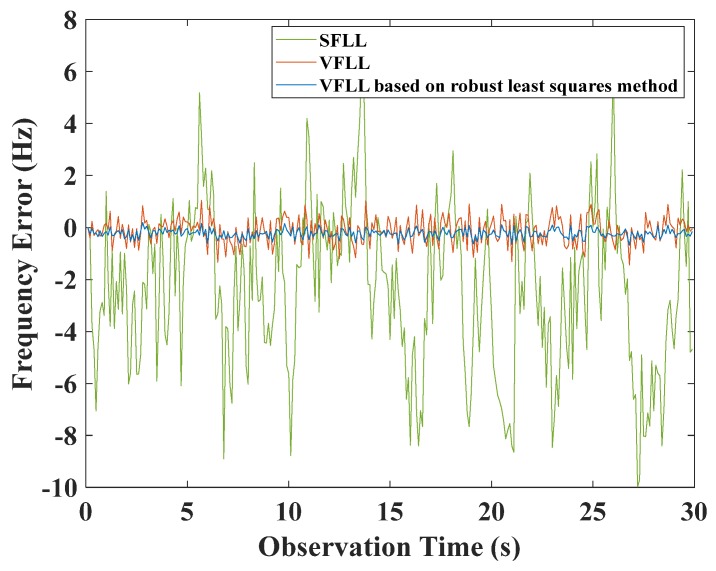
The frequency tracking error of channel CH6.

**Figure 13 sensors-20-01224-f013:**
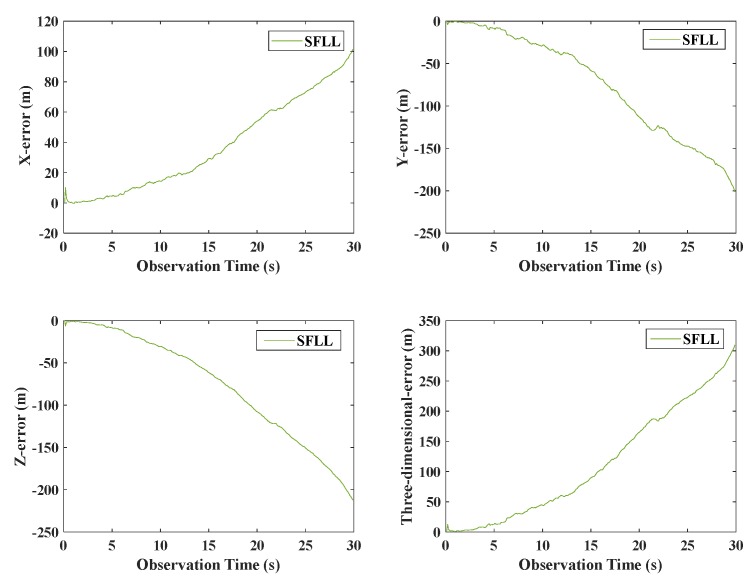
The overall positioning error of the traditional SFLL model.

**Figure 14 sensors-20-01224-f014:**
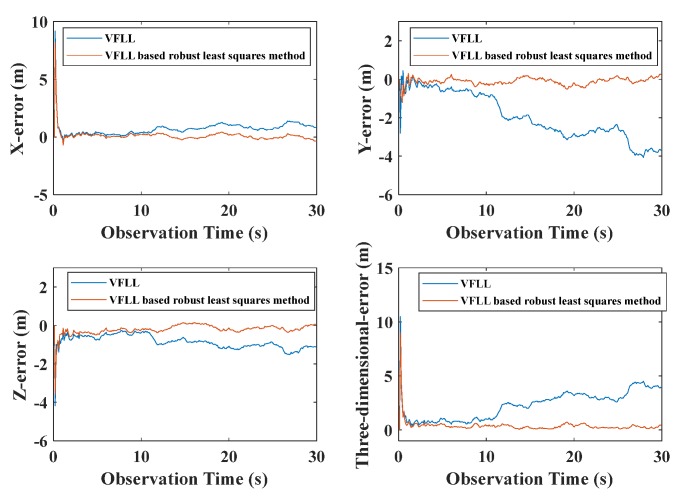
The overall positioning errors of the two VFLL models.

**Figure 15 sensors-20-01224-f015:**
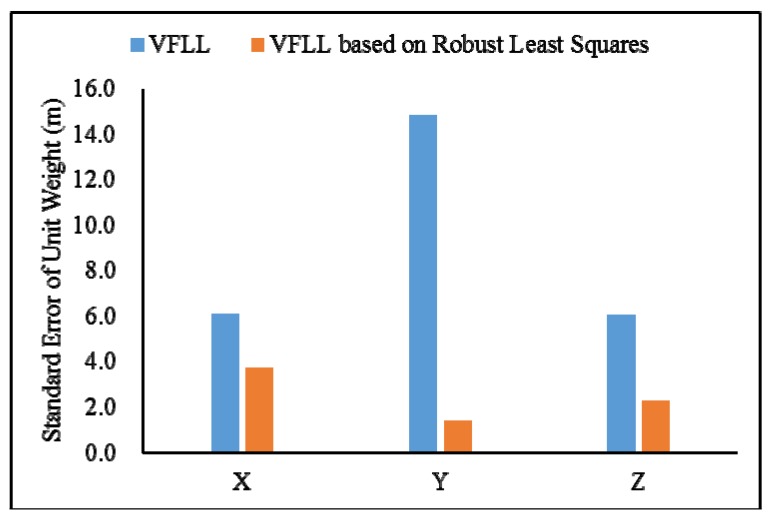
The standard error of unit weight of positioning error of two model.

**Figure 16 sensors-20-01224-f016:**
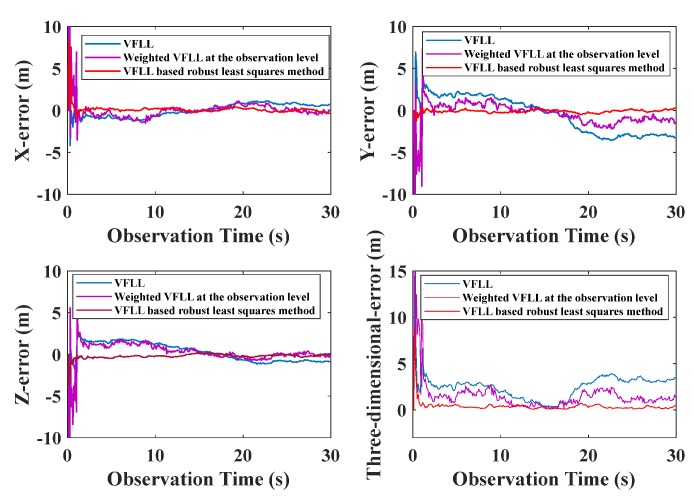
Positioning accuracy of the three models.

**Figure 17 sensors-20-01224-f017:**
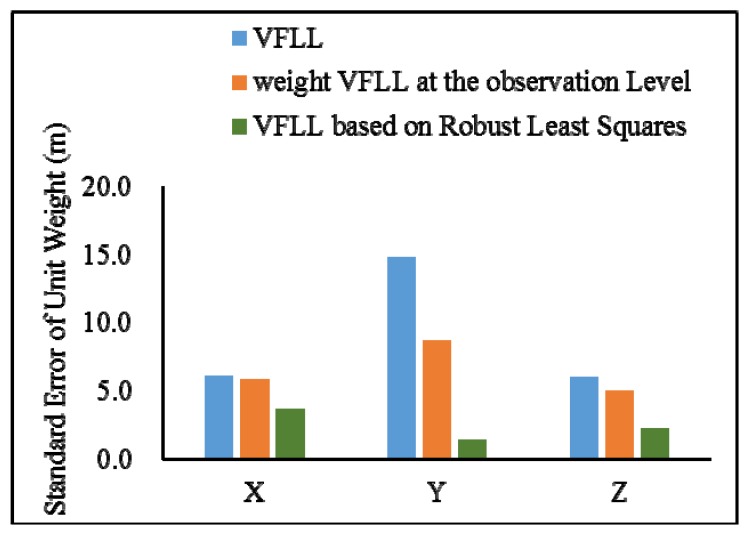
The standard error of a unit weight of the positioning error of three models.

**Table 1 sensors-20-01224-t001:** The standard deviation of frequency error of all the channels.

	SFLL	VFLL	VFLL Based on the Robust Least Squares Method
	Frequency Error (Hz)	Frequency Error (Hz)	Frequency Error (Hz)
CH1	2.51	0.45	0.22
CH2	2.14	0.48	0.23
CH3	1.87	0.44	0.21
CH4	1.97	0.41	0.20
CH5	2.67	0.45	0.22
CH6	1.89	0.49	0.23
CH7	2.07	0.47	0.23
CH8	1.79	0.47	0.23
CH9	2.44	0.47	0.21
CH10	2.34	0.45	0.21
CH11	1.96	0.46	0.22

**Table 2 sensors-20-01224-t002:** The standard deviation of the velocity error.

	*Vx* (m·s^−1^)	*Vy* (m·s^−1^)	*Vz* (m·s^−1^)
SFLL	0.440	0.644	0.379
VFLL	0.103	0.130	0.083
VFLL based on the robust least-squares	0.049	0.062	0.040

**Table 3 sensors-20-01224-t003:** The standard deviation of the frequency tracking error of all the channels.

	SFLL	VFLL	VFLL Based on the Robust Least Squares Method
	Frequency Error (Hz)	Frequency Error (Hz)	Frequency Error (Hz)
CH1	2.12	0.36	0.35
CH2	1.75	0.39	0.37
CH3	1.64	0.36	0.35
CH4	1.60	0.35	0.34
CH5	3.62	0.40	0.38
CH6	3.07	0.50	0.17
CH7	3.06	0.50	0.17
CH8	2.76	0.49	0.17
CH9	3.85	0.50	0.18
CH10	50.30	2.00	/
CH11	36.70	2.15	/

**Table 4 sensors-20-01224-t004:** The standard deviation of the positioning error of the three models.

Model	X (m)	Y (m)	Z (m)	Three-Dimensional Positioning Error (m)
Traditional SFLL	30.09	60.09	62.43	91.66
Traditional VFLL	0.35	1.05	0.47	1.17
Proposed VFLL	0.02	0.26	0.07	0.27

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
