# Peer review of "The Application of Robust Least Squares Method in Frequency Lock Loop Fusion for Global Navigation Satellite System Receivers"

_sensors, 2020, doi:10.3390/s20041224_

Round 1

Reviewer 1 Report

Review of “The Application of Robust Least Squares Method in FLL Fusion for GNSS Receivers” by Han et al.

This is an interesting contribution on an important GNSS problem. There are however a number of improvements necessary before the manuscript can be accepted for publication. These are listed below:

The authors speak of robust least-squares methods but do not specify what they mean by robust, robust against what? This should be clarified. The authors give in their introduction a too limited overview of robust least-squares method. Those that make use of statistical testing are not mentioned. For instance the DIA-estimator of [Teunissen (2018): Distributional theory for the DIA method. Journal of Geodesy, 92, pages59–80(2018)] is also a robust least squares method. The authors should therefore give a better overview of such methods. The authors mention ‘Krarup and Kubil’, but this should be ‘Krarup and Kubik’ With respect to the Danish method the authors refer to [8]. But this is not the original publication where the method was introduced. The authors should refer to the original publication of the Danish method. Below Eq(8) the authors refer to sigma as the mean square error. This is incorrect. Therefore replace sigma with sigma-squared.

Author Response

Response to Reviewer 1 Comments

Point 1: The authors speak of robust least-squares methods but do not specify what they mean by robust, robust against what?

Response 1: Thanks for your advice. The robust feature means maintaining good stability and ability to resist interference signal shelter and interference. And I have added a detailed explanation in the third paragraph of the introduction.

Point 2: The authors give in their introduction a too limited overview of robust least-squares method. Those that make use of statistical testing are not mentioned. For instance the DIA-estimator of [Teunissen (2018): Distributional theory for the DIA method. Journal of Geodesy, 92, pages59–80(2018)] is also a robust least squares method. The authors should therefore give a better overview of such methods.

Response 2: Thanks for your advice. I have carefully read the relevant literature you recommended named Distributional theory for the DIA method. And summarized other related data statistical testing methods in the fourth paragraph of introduction and the reference [14],[15] and [16] are about them.

Point 3: The authors mention ‘Krarup and Kubil’, but this should be ‘Krarup and Kubik’ With respect to the Danish method the authors refer to [8]. But this is not the original publication where the method was introduced. The authors should refer to the original publication of the Danish method.

Response 3: Thanks for your advice. I'm sorry for the wrong name but I have corrected it in the manuscript. And for the reference [8], I have changed it to the original publication where the Danish method was introduced.

Point 4: Below Eq(8) the authors refer to sigma as the mean square error. This is incorrect. Therefore replace sigma with sigma-squared.

Response 4: Thanks for your advice. I'm sorry I didn't find sigma under below (8), could it be (28) in the original manuscript? There were some errors in the equation (28) for the original manuscript, so I corrected them. And the equation (28) was changed to (29). About the sigma, it represents the standard deviation of the ith residual and I also corrected it.

Thanks again for all your comments!

Reviewer 2 Report

This paper proposed the fusion of VLL and robust weighted least squares (WLS) technique to obtain more accurate frequency estimation/tracking. Extensive experimental results corroborated the validity of the proposed technique. This paper appears to have some merits. But in my opinion, a major revision is required before it could be considered appropriate for publication.

Major comments are in order.

1). The English language usage in this manuscript is below average. Significant revision efforts are needed to improve the clarity of the presentation. Some examples of awkward expressions and grammatical mistakes are

a). In Abstract, "application prospect"

b). In Introduction, page 1, "in military and civil, higher ..."

c). In Introduction, page 1, "the FLL and robust least-squares fusion ..."

d). In Introduction, page 2, "The vector tracking algorithm first models ... and then realize..."

e). In Introduction, page 2, "used to locate can be estimated ..."

f). In Introduction, page 2, "...for robust estimation of data processing ..."

g). In Introduction, page 3, "the specific ranges of high and low SNR..."

h). In Section 2, page 3, "to keep the frequency between the duplicated carrier and the receiver carrier..."

2). In my opinion, a diagram of the proposed scheme would be quite helpful to clarify the structure of the developed VFLL.

3). In Eqn. (3), T should be T_L.

4). To derive Eqns. (10) and (11), did the authors simply combining (7) and (9) as well as (8) and (9) and solve for SNR/T?

5). On page 8, what did the authors mean by saying "At a high SNR, R_0^n satisfies the Gaussian distribution of 2/(SNR*T_L^2)"?

6). The sentence above (20) is quite awkward and hard to follow.

7). When deriving (22), it would be interesting to provide details on how to obtain the initial values for \alpha, \tau and \phi.

8). Under (24), what is 'the satellite weight'?

9). It was claimed that a Kalman filter (KF) was adopted for Doppler frequency estimation. This sound strange to me. The authors attempted to improve the performance of VLL. Now, they considered Doppler frequency tracking. Please elaborate on this point.

10). In (26), -T/F*I_{M*M} should be -T*c/F*I_{M*M}, where c is the speed of light (radio propagation speed).

11). The estimated \delta\theta was considered process noise in (26). The definitions of R and \bar{P}_n were not given. These make Section 3.2 almost unreadable.

The connection between the weighting matrix and the segmentation of SNR regions is quite vague. The filtering process should at least be described through providing an algorithm flow. 

12). The two paragraphs under (29) are difficult to understand. 

Author Response

Response to Reviewer 2 Comments

Point 1: The English language usage in this manuscript is below average. Significant revision efforts are needed to improve the clarity of the presentation. 

Response 1: Thanks for your advice. For this problem, we used professional English editing service to improve it. Some examples of awkward expressions and grammatical mistakes that you point out have been corrected carefully.

Point 2: In my opinion, a diagram of the proposed scheme would be quite helpful to clarify the structure of the developed VFLL

Response 2: Thanks for your advice. So, a diagram in the Figure 2. is added to illustrate the structure of the proposed model. In the Figure 2, each channel is used to track a satellite, and δα, δτ, and δφ respectively denote differences in the signal amplitude, code phase, and carrier phase between the replicated carrier and the received carrier. The least square method is used to calculate the values of δα, δτ and δφ. Then, these values are used to calculate precisely the carrier frequency difference, δf, by the Kalman filter. In the Kalman filter, the equivalent weight matrix is constructed by Equation (28),

Point 3: In Eqn. (3), T should be T_L

Response 3: Thanks for your advice. So I have corrected it in the equation (3).

Point 4: To derive Eqns. (10) and (11), did the authors simply combining (7) and (9) as well as (8) and (9) and solve for SNR/T?

Response 4: Yes, it is. Because the equation (9) represents the numerical solution to the output variance of a discriminator in all SNR, the equation (7) represents the variance at high SNR, so combine the (7) with (9) to determine the specific value of high SNR in order to get the equation (11). Similarly, the equation (8) represents the variance at low SNR. So combine the (8) with (9) to determine the specific value of low SNR in order to get the equation (10). The value of SNR is obtained by the combing, and then the equation (CNR=SNR/T) is used to convert SNR into CNR.

Point 5: On page 8, what did the authors mean by saying "At a high SNR, R_0^n satisfies the Gaussian distribution of 2/(SNR*T_L^2)"?

Response 5: Because the   represents the total discriminator noise power, so it is the  in the equation (4). Therefore, the  satisfies the Gaussian distribution at high SNR, and the variance is the . So, it can be substituted into equation (12) to calculate the specific value of high SNR after filtering.

Point 6: The sentence above (20) is quite awkward and hard to follow.

Response 6: Thanks for your advice. I'm sorry I didn't express the meaning clearly. So I have corrected it and hope to make it clear. This is the basic definition of baseband processing in a receiver and I also made corresponding changes in the article.

The downlink signal in the satellite navigation receivers represents an input to the baseband module after frequency conversion processed by the radio frequency (RF) module which is expressed by equation (20).

In the equation (20) , the  denotes the signal receiving time,  represents the number of received satellite, is the signal amplitude,  is the code phase delay, is the carrier phase and  is the carrier frequency, respectively;  denotes the modulation code of the corresponding signal, and  represents the white Gaussian noise.

The process of receiver tracking is to obtain the above-mentioned signal parameters from the noisy signal which are , , and . In the traditional loop, the parameters were obtained by the independent discriminator. However, in this model, the parameters were obtained by the least squares method and Kalman filtering.

Point 7:  When deriving (22), it would be interesting to provide details on how to obtain the initial values for \alpha, \tau and \phi.

Response 7: We linearize (20) by the Taylor expansion and we get the (21). But (21) represents the linearization of ith channel at one point. Because the receiver for GNSS has multiple channels for tracking at the same time, the (21) is expressed as a matrix by the (22). In iterative computation, the initial values of δα, δτ and δφ was set to zero. Through time increment, iterative calculation is carried out to finally calculate the residual by the least squares.

Point 8:  Under (24), what is 'the satellite weight'?

Response 8: Thanks for your advice and I'm sorry I didn't express the meaning clearly. In GNSS receiver, each channel is responsible for tracking a satellite. The meaning is that weight of each channel in obtaining the frequency value of  .

Point 9:  It was claimed that a Kalman filter (KF) was adopted for Doppler frequency estimation. This sound strange to me. The authors attempted to improve the performance of VLL. Now, they considered Doppler frequency tracking. Please elaborate on this point.

Response 9: Thanks for your advice and I'm sorry I didn't express the meaning clearly. This is my misrepresentation, and to be more precise, it should be the carrier frequency. The GNSS receiver tracks the signal mainly by means of carrier tracking loop and code tracking loop. What’s more, the VFLL is a form of carrier tracking loop and it is  to keep the same frequency between the replicated duplicated carrier and the received carrier. So the VFLL and Kalman filter (KF) was adopted for carrier frequency. At the same time, more accurate and robust frequency observations can be obtained to improve the performance of the VFLL

Point 10: 10). In (26), -T/F*I_{M*M} should be -T*c/F*I_{M*M}, where c is the speed of light (radio propagation speed).

Response 10: Thanks for your advice. So I have corrected it in the equation (26).

Point 11: The estimated \delta\theta was considered process noise in (26). The definitions of R and \bar{P}_n were not given. These make Section 3.2 almost unreadable.

The connection between the weighting matrix and the segmentation of SNR regions is quite vague. The filtering process should at least be described through providing an algorithm flow.

Response 11: Thanks for your advice. The R denotes the noise covariance matrix and it is represented the R=(ATA)-1 this KF model. The denotes the product of the weight factor matrix ( ) and the initial weight matrix.

In this model, the initial weight matrix  is defined as the identity matrix, and the weight factor is obtained by the three-segment weight function in equation (31). As each observation time increases, the weight matrix is constantly updated.

The specific range of high and low SNR of the VFLL is determined by the segmentation results and the results are the basis for the weight selection. The expression in Equation (29) requires a known residual value, so it can be used only in the data processing stage. So, in order to determine the weights at the signal level, we need to define a weight function according to the segmentation results.

And according to your advice, I added figure 3 to illustrate the filter algorithm flow in detail. In the Figure 3, at time k, the δα, δτ and δφ are the inputs to the Kalman filter, the  and R are used to calculate the gain matrix in the Equation (27). The   calculated by the Equations (28) and (31) is the equivalent weight matrix that relates to the SNR. Then, the δf was obtained. And then, at time k+1, the Kalman filter repeats all these steps.

Point 12: The two paragraphs under (29) are difficult to understand.

Response12: Thanks for your advice and I'm sorry I didn't express the meaning clearly. Since an equation is added, the (29) in the original manuscript is (31) now. I rephrased the meaning under (31). The represents the weight factor of ith channel and it can form a diagonal matrix which was weight factor matrix ( ) used to obtain the equivalent weight matrix ( ). Thus, when the values in Equations (27) and (26) are known, the value of carrier frequency can be calculated.

Thanks again for all your comments!

Reviewer 3 Report

The manuscript proposed a new tracking loop combining the VFLL with robust least squares algorithm to ensure the tracking accuracy of GNSS receivers. The manuscript has the necessary content and the structure and English language is clear. I suggest this paper to be published after a minor revision. The detailed comments are presented as follows:

1) In Sect. 2, there is one citation about the algorithm for single point positioning with application of the least squares method to determine the positioning parameters and adopted the regularization method, which has not any relation to the topic of this manuscript. It is suggested that this citation should be removed.

2) In Sect. 3, in the observation equation in the matrix form (23), the design matrix in the Least Squares Method is usually written by A, which is more clear for worldwide readers.In the observation equation (24) the reduced observation vector Δx should be defined! More specifically, there is two errors in the cited IGGIII model (28), which should be read as:

      1  | vi| ≤ k0

yi = k0/|vf,if,i|*((k1-|vf,if,i|)/(k1-k0))2 k0 < | vi| ≤ k1

        0   | vi| > k1

The related weight function (30) should be corrected accordingly. Please check the right reference of IGGIII, for example by early published book (Robustified Least Squares Approaches, in Chinese) by Zhou, Huang, Yang and Ou (1995, p. 118). Otherwise, the author misunderstands the meaning of mean square error and standard deviation σf,i denotes the standard deviation here.

Therefore, these examples calculated with these two formulas should be checked and revised.

In addition, in order to consider the robustness in structure space, the standardized residual should be used:

vi = vi/σvi

In which the σvi represnets the standard deviation of the ith residual, rather than the priori standard deviation of the ith observation (σf,i  ).

3) In Sect. 4, all the data and graphs only show the errors of the observed data such as frequency and velocity, but do not show the estimated unknown parameters with different methods. The results are not clear if these estimation methods have real effects on the parameters. In addition, please give more statistical data like root mean square residual and standard deviation of unit weight for each method. These data can intuitively compare the results of various methods from a statistical perspective.

Author Response

Response to Reviewer 3 Comments

Point 1: In Sect. 2, there is one citation about the algorithm for single point positioning with application of the least squares method to determine the positioning parameters and adopted the regularization method, which has not any relation to the topic of this manuscript. It is suggested that this citation should be removed.

Response 1: Thanks for your advice. I have deleted the references here according to your advice.

Point 2: In Sect. 3, in the observation equation in the matrix form (23), the design matrix in the Least Squares Method is usually written by A, which is more clear for worldwide readers.In the observation equation (24) the reduced observation vector should be defined!More specifically, there is two errors in the cited IGGIII model (28), which should be read as:

The related weight function (30) should be corrected accordingly. Please check the right reference of IGGIII, for example by early published book (Robustified Least Squares Approaches, in Chinese) by Zhou, Huang, Yang and Ou (1995, p. 118). Otherwise, the author misunderstands the meaning of mean square error and standard deviation. σ_(f,i) denotes the standard deviation here.

Therefore, these examples calculated with these two formulas should be checked and revised.

In addition, in order to consider the robustness in structure space, the standardized residual should be used:

In which the    represnets the standard deviation of the ith residual, rather than the priori standard deviation of the ith observation ( ).

Response 2: Thanks for your advice. Your comments are very accurate and detailed. In the equation (23), the least squares coefficient matrix has been represented by A.

In the equation (24), The   denotes the difference between  and  .According to your comments, I have defined it in this article under (24).

According to your advice, I have corrected two errors in the IGGIII model in the (28) in the original manuscript, but now it’s (29), and added the equation (30).

Point 3: In Sect. 4, all the data and graphs only show the errors of the observed data such as frequency and velocity, but do not show the estimated unknown parameters with different methods. The results are not clear if these estimation methods have real effects on the parameters. In addition, please give more statistical data like root mean square residual and standard deviation of unit weight for each method. These data can intuitively compare the results of various methods from a statistical perspective.

Response 3: Thanks for your advice. I'm sorry I didn't make the meaning clear. Rather, the frequency error in all figures and tables should be the frequency residual. It represents the estimated unknown parameter,  in the (20). Because the robust least squares is for VFLL, only the unknown coefficient  has been effected. What’s more, the velocity and the positioning accuracy are also the parameters. And they have been analyzed in the article.

According to your advice, I used the unit weight for each method to compare the results of various methods from a statistical perspective. They are shown in the figures 8,15 and 17.                                                               

Thanks again for all your comments!
